# Assessing adolescents' critical health literacy: How is trust in government leadership associated with knowledge of COVID-19?

Channing J. Mathews[1]*, Luke McGuire[2‡], Angelina Joy[1‡], Fidelia Law[2‡], Mark Winterbottom[3‡], Adam Rutland[2‡], Marc Drews[4‡], Adam J. Hoffman[5‡], Kelly Lynn Mulvey[1], Adam Hartstone-Rose[6]

1 Department of Psychology, North Carolina State University, Raleigh, North Carolina, United States of America, 2 Department of Psychology, University of Exeter, Exeter, United Kingdom, 3 Faculty of Education, University of Cambridge, Cambridge, United Kingdom, 4 EdVenture, Columbia, South Carolina, United States of America, 5 Department of Psychology, Cornell University, Ithaca, New York, United States of America, 6 Department of Biological Sciences, North Carolina State University, Raleigh, North Carolina, United States of America

☉ These authors contributed equally to this work.
‡ These authors also contributed equally to this work.
* cjmathew@ncsu.edu

## Abstract

This study explored relations between COVID-19 news source, trust in COVID-19 information source, and COVID-19 health literacy in 194 STEM-oriented adolescents and young adults from the US and the UK. Analyses suggest that adolescents use both traditional news (e.g., TV or newspapers) and social media news to acquire information about COVID-19 and have average levels of COVID-19 health literacy. Hierarchical linear regression analyses suggest that the association between traditional news media and COVID-19 health literacy depends on participants' level of trust in their government leader. For youth in both the US and the UK who used traditional media for information about COVID-19 and who have higher trust in their respective government leader (i.e., former US President Donald Trump and UK Prime Minister Boris Johnson) had lower COVID-19 health literacy. Results highlight how youth are learning about the pandemic and the importance of not only considering their information source, but also their levels of trust in their government leaders.

## Introduction

On February 28[th], 2020, former US President Donald Trump received criticism for his politicization of the coronavirus pandemic by referring to it as "a Democratic hoax" [1]. The implications of his suggestion polarized both the country and news media, as the public scrambled to disentangle fact from fiction [2]. On March 3[rd], 2020, UK Prime Minister Boris Johnson announced, "I'm shaking hands continuously. I was at a hospital the other night where I think there were actually a few coronavirus patients and I shook hands with everybody, you'll be pleased to know. I continue to shake hands" [3]. On the same day, the UK Scientific Advisory

**Data Availability Statement:** All relevant data are deposited within the Open Science Framework at the following URL: https://osf.io/nj3bf/.

**Funding:** This work was supported in the United States by the National Science Foundation AHR, KLM, AR [Grant Number: DRL-1831593]; and collaboratively in the United Kingdom by the Wellcome Trust [Grant Number: 206259/Z/17/Z] and the Economic and Social Research Council. Opinions, findings, and conclusions from this report are those of the authors and do not necessarily reflect the views of the National Science Foundation or the Wellcome Trust/Economic and Social Research Council. The funders had no rle in study design, data collection and analysis, decision to publish, or preparation of this manuscript.

**Competing interests:** The authors have declared that no competing interests exist.

Group for Emergencies released recommendations that the "Government should advise against greetings such as shaking hands and hugging, given existing evidence about the importance of hand hygiene. A public message against shaking hands has additional value as a signal about the importance of hand hygiene" [4]. Further, rumors and conspiracy theories have exacerbated misinformation about the COVID-19 pandemic by calling scientific facts into question.

The US and the UK have become epicenters of the COVID-19 pandemic [5] and navigating different sources of information about the pandemic has proved challenging due to the proliferation of inaccurate reporting and misinformation [6–9], including inaccurate guidance espoused by government leaders. The ambiguity between fact and fiction has dangerous public health implications [8, 10], particularly among adolescent youth as they are especially vulnerable to misinformation [11]. The current study aims to examine the health literacy of STEM oriented adolescents (i.e., adolescents involved in STEM-focused extracurricular educational programs) around COVID-19, with attention to sources of news information and trust as key factors that may shape what adolescents know and believe about COVID-19.

The rates of spread of COVID-19 in the US and in the UK have been associated with the slow response of both governments to implement social distancing and lockdown guidelines [12, 13]. Further, given the stay at home and social distancing orders required by the COVID-19 pandemic, people are likely more reliant on news media as a way to stay informed as the pandemic progresses [14–16]. However, given the sociopolitical moment, the credibility of the news landscape and its ability to disseminate accurate knowledge of COVID-19 has been called into question. This news is particularly consequential for adolescents, who are using online media as means of both staying informed and for social connection [17, 18]. Very little is known about what adolescents know and believe about viruses, and COVID-19 in particular. Some research, however, suggests that adolescents' knowledge of infectious disease and viruses, in general, is quite limited [19]. Scant research has examined what adolescents know about COVID-19, although one investigation of Jordanian adolescents suggests that their understanding of COVID-19 (i.e., transmission, prevention, and control of the virus' spread), in particular, was fairly accurate [20]. However, this study was specific to adolescents' COVID-19 knowledge and attitudes toward *preventative* measures toward COVID-19, and did not assess broader knowledge of the science of viruses and more comprehensive COVID-19 health literacy. Knowledge measures were a series of true/false responses, which have a higher chance for accurate responses (50%), than in the case of a traditional four response multiple choice question (25%). As our understanding of adolescents' health literacy is limited, more research is needed to expand our understanding of adolescents' knowledge, and of factors associated with more accurate knowledge around COVID-19.

Given that accurate information and critical consumption of COVID-19 news media is associated with better preventative behaviors [21, 22], it is important to examine how COVID-19 news consumption impacts teens' COVID-19 health literacy. Research with adults in the US suggests that trust in government sources is associated with greater COVID-19 health literacy, while trust in online and social media sources was associated with less COVID-19 health literacy [14]. In the present study, we investigate how STEM oriented adolescents' news sources (traditional or social media news) are associated with their COVID-19 health literacy. Further, we investigate how adolescents' news consumption and their trust in the COVID-19 information shared by different sources (including government leaders, different types of news sources, and scientists) is associated with adolescents' COVID-19 health literacy.

## The health literacy skills framework

This study draws upon the Health Literacy Skills Framework (HLS) [23] as a conceptual model of the associations between adolescents' news consumption, trust, and COVID-19 health literacy. The HLS acknowledges the influence of ecological factors on the ways individuals consume and apply health-related information. Specifically, this framework identifies factors such as media and credibility of information source (e.g., perceptions of how trustworthy a source is) are critical to how health information is processed by the consumer. Further, this model acknowledges how demographic factors, such as gender, and contextual factors such as prior knowledge, play a role in understanding a novel health concern, such as the spread of the COVID-19 virus. Adolescence is a developmental period characterized by meaningful change, including continued development of executive functions related to decision-making [24, 25]. However, given that cognitive processes related to planning and decision-making lag behind other development features of adolescence, it may be difficult for youth to meaningfully engage with health-related news information, particularly in the case of the novel coronavirus. As health information related to the COVID-19 virus is constantly shifting, adolescents and young adults may have difficulty identifying and evaluating credible information sources. Yet, young people who have had increased exposure to STEM contexts, may be more equipped to navigate various information sources given their prior experience with consuming STEM related information.

## Adolescents' news consumption

Although recent reports suggest that adolescents' consumption of news media is generally on the decline, other findings suggest that teens have simply transitioned their news media consumption from print to digital media [26]. Indeed, adolescents and young adults are more likely to access news via social media platforms such as Facebook, Twitter and Instagram as such platforms allow them to be actively engaged in both consuming and sharing information [27, 28] Yet, social media encourages heuristic information processing (i.e., shortcut thinking), which may be associated with adolescents having lower capacity to identify and evaluate factual knowledge about COVID-19 [29]. In addition, research suggests that adolescents have difficulty distinguishing between real and fake news, which can contribute to the spread of COVID-19 misinformation [11, 29, 30]. However, youth are often aware and skeptical of personalized social media algorithms, leading them to seek a variety of news sources to triangulate credibility and engage alternative viewpoints [28, 30]. Thus, though youth may not be engaging with print and online news platforms at the same rates as social media, they may use non-social media sources as fact checkers given that they trust established news organizations more than the social media news "snacks" that are pushed on their timelines and newsfeeds [28, 30].

With respect to the COVID-19 pandemic, youth's engagement with multiple sources of news has important health implications. Riiser and colleagues' [31] study of Norwegian adolescents found that youth's engagement with traditional news media was associated with greater awareness and practice of precautionary measures around COVID-19. Such engagement seems to strengthen youth's understanding of the severity and transmissibility of the virus, leading them to be more socially responsible in community efforts to combat the pandemic [18]. Given the association between news engagement and preventative behaviors to prevent the spread of the COVID-19 virus and the likelihood of young people engaging multiple news sources, it is important to investigate the role of news media in adolescents' and young adults' knowledge of COVID-19.

## The role of trust in adolescents' information seeking behaviors

Trust is a fundamental component of how individuals access and comprehend knowledge, particularly in scientific and health domains [32, 33]. Societies depend on the expertise of scientists and public health experts to help them navigate and process new information, and such dependence requires an individual's trust in various experts' knowledge and scientific skill sets. For adolescents, this trust may be encompassed within their immediate networks (e.g., peers, family), proximal networks (e.g., teachers, scientists) and by public domains (e.g., federal government, public health organizations). Specific to COVID-19 related information, family and peers may play a role in the sharing of information and are likely to be deemed credible sources given their proximity and direct influence on teens' lives [3, 4]. Although family and peers may be deemed more trustworthy due to personal relationships, this trust does not guarantee that youth access the most credible resources. Family and peers are critical in sharing information but may not have the skills to evaluate what information or resources are the most trustworthy, particularly as new science emerges.

As an alternative to family or peers, adolescents may turn to teachers or public scientists as trustworthy knowledge sources given their roles in the teaching and dissemination of novel information [34, 35]. Public opinion surveys suggest that Americans' trust in science and scientists have remained relatively stable over time with variation across political party, religious views, and rural vs. urban contexts [35, 36]. However, a study analyzing the impact of epidemics on trust in scientists suggests that though trust in science remains stable after an epidemic experience, trust in individual scientists has a sharp decline during and after an epidemic's onset [37]. Although these findings may initially seem discrepant, it may be that information disseminated by public health institutions, such as the World Health Organization (WHO) or the Center for Disease Control (CDC), may be seen as more trustworthy because these organizations are perceived as less susceptible to political bias or scrutiny [37, 38]. Given the role and visibility of institutions and individual scientists, such as Dr. Anthony Fauci in the US, in the public health messaging to slow the spread of the COVID-19 virus, it is critical to explore how trust in these individuals and groups matter in the development of one's knowledge about COVID-19.

Beyond scientific institutions, government leaders may play a critical role in the messaging and modeling of healthy hygiene behaviors in the midst of a public health crisis [20, 39]. Despite its importance, research suggests that trust in government leaders is on the decline, due to events that reveal corruption or undermine government legitimacy [39–41]. Further, given the lack of government responsiveness and transparency during the COVID-19 pandemic, it may be beneficial for individuals to take a cautious approach when considering how to interpret information from government officials. Government responsiveness and transparency are considered key factors in garnering public trust, as both can be considered consequences of direct democracy (i.e., ordinary individuals' influence on policy and practice [42].

Unfortunately, governments do not always exhibit both high transparency and high responsiveness. Moreover, specifically considering the US and UK contexts, various scandals have affected how the public has viewed each government's effectiveness in responding to COVID-19 [43, 44]. For example, in the UK, the Prime Minister's aid, Dominic Cummings faced widespread public and media criticism for violating UK lockdown protocols. His disregard of COVID-19 public safety measures, coupled with the Prime Minister's failure to reprimand Cummings, was associated with a decline in government trust in addressing the severity of the pandemic [44]. In the US, former President Trump's consistent downplay of the severity of the COVID-19 virus aligned with his apparent efforts to undermine scientific initiatives to secure public health. For example, Former President Trump eliminated the PREDICT organization, a

US funded program responsible for anticipating and containing potential pandemics, two months prior to the arrival of COVID-19 on US shores [43]. At the height of the pandemic, Former President Trump consistently undermined the public health recommendations of Dr. Anthony Fauci, the US director of the National Institute of Allergies and Infectious Diseases, including criticizing Fauci for his emphasis on continued social distancing, quarantine lock-downs, and more scientific testing for COVID-19 drug treatments [45, 46]. In light of confusing and, at times, misleading information shared by government officials during the pandemic, it may be that trust in government officials plays a role in how much adolescents learn about the COVID-19 virus and how accurate their knowledge is. Thus, we expect to observe that. . .

## Adolescents' public health literacy and knowledge of COVID-19

Health literacy, an individual's ability to seek, analyze, and evaluate credible health information, and use it to inform their health-related behaviors [47–49], is a key factor associated with individuals' prosocial public health behaviors, particularly around vaccination [49]. However, little work has explored adolescent and young adult health literacy, particularly within a STEM-engaged sample. Adolescent health literacy is critical during the COVID-19 pandemic as young people may transmit the virus to more vulnerable populations such as the elderly [50]. However, several barriers exist for youth to access and interpret credible information in a continuously shifting COVID-19 knowledge base [2, 49, 51]. Whereas youth express an interest to stay informed about health-related behaviors, basic accessibility to such information is somewhat limited. In a study of adolescents' health literacy in the US and the UK, Gray and colleagues [51] highlighted the ways that youth struggled to access information due to difficulties in spelling medical terms of interest, lack of clarity around how to evaluate credible resources, or a meaningful strategy to sift through multitudes of information brought up by a relevant search engine. Yet, youth in the UK were noted to be at an advantage in evaluating the credibility of health-related information due to the National Health Service, a publicly funded centralized health system in the UK. In contrast, youth in the US could not identify a similar resource and instead turned to broad media platforms, such as Encyclopedia Britannica, to access health information [51]. Yet, with advances in technology and technology literacy among adolescents, youth today may be better equipped to evaluate various forms of health information, particularly if they have practical experiences engaging in STEM focused education [52]. Further, there has been a sharp increase in youth using online sources for news related to science and technology, suggesting that it has become a more mainstream source of information since the Gray and colleagues study [36, 50]. Continued research to examine where adolescents obtain health information, in particular related to COVID-19, is warranted given the limited prior research in this area.

Beyond the mechanics of identifying credible health information, research suggests that adolescents have strong misconceptions regarding how viruses work [19, 53]. In an intervention study, Dumais and Hasni [19] demonstrated that high school biology students lacked both the scientific knowledge and vocabulary to describe the structure and transmission of viruses. In a subsequent study of both secondary and university students' virus knowledge, Simon and colleagues [53] noted that while viral knowledge and interest increased with age, such knowledge was fragmented at best, particularly in the misidentification of a virus as a bacterium, which have different treatment pathways (i.e., vaccine versus antibiotic treatment). Such evidence suggests that school science curricula may not bolster the health literacy of youth and young adults, leaving them more dependent on other information avenues such as online or news media. This association with media may be particularly true in the case of virus

transmission that leads to pandemics. One particular virus that has obtained much research attention is HIV. Studies of youth knowledge and attitudes towards HIV, including their ability to critically evaluate myths and facts about viral transmission, show that such information was largely drawn from mass media rather than school curriculum [54–56]. Yet, the challenges of navigating massive amounts of information, coupled with identifying what information is actually credible, create a formidable barrier to youth's health literacy that has implications for their health-related behaviors [2]. Thus understanding the interplay between youth's sources of virus knowledge and their level of health literacy can help us understand how youth adhere or fail to adhere to prosocial health behaviors.

In addition to adolescents' limited knowledge of viruses, barriers to health literacy undermine youth's need for autonomy and desire to make informed decisions for themselves. These barriers are further complicated by the fact that adolescents are still undergoing development in judgment and decision-making, and executive function abilities across into early adulthood which can impact their choices and decisions [24, 25]. In the midst of the COVID-19 pandemic, the consequences of some of these decisions may be deadly given the high level of transmissibility of the virus, particularly among young people [50] and ensuring that adolescents have accurate, up-to-date knowledge about COVID-19 may inform better decision making. Adolescents and young adults have been involved in superspreader events—large social gatherings where a few individuals can infect many others—which can prove deadly for the older adults to whom they expose COVID-19 virus at home [57]. American holiday parties, such as the 4th of the July and spring break travel have garnered national attention as thousands of youth gathered together without abiding to social distancing recommendations from the CDC [58, 59]. Health professionals in the UK have identified the role of older adolescents and young adults in the upticks in COVID-19 infection cases as a result of house parties and music events that youth attend [60]. Given the consequences of ignoring public health guidelines, it is critical for researchers to both establish how youth are obtaining knowledge about COVID-19, and if youth can distinguish fact from fiction regarding exposure and transmission of the virus.

## The current study

Given the lack of research on young peoples' knowledge of COVID-19 and the factors that influence such knowledge, the present study examines how different forms of news media are associated with youth knowledge of COVID-19 in a sample of adolescents and young adults that are involved in adolescent programming at six informal STEM learning sites (museums, zoo, etc.) in the US and UK. Further, given the potential roles of news media, teachers, scientists, public health organizations (e.g., the WHO), and government leaders around the public health messaging of the pandemic, we investigate how trust in these information sources may change the relationship between news source and knowledge of COVID-19. Aligned with the HLS framework [23], we expected that youth who obtained their news from traditional media sites (i.e., online and print news) would have higher COVID-19 health literacy. In contrast, youth who sought their news from social media sites (e.g., Twitter, Facebook) would have lower COVID-19 health literacy. We expected that trust in COVID-19 information from teachers, traditional media, scientists, and the WHO would be associated with better performance on a COVID-19 assessment, as these sources are likely sharing accurate COVID-19 information. Further, trust in information related to COVID-19 shared by government leaders, the government in general, and social media might be associated with lower performance on the assessment, given mixed messages around COVID-19 that have been communicated by these sources. Additionally, we expected that trust in teachers, traditional media, scientists,

and the WHO would moderate the association between COVID-19 news source and health literacy such that higher trust in these information sources would strengthen the relationship between news source and COVID-19 knowledge. Finally, we expected that greater trust in the information shared by the government, government officials and social media would also moderate the association between COVID-19 news source and factual knowledge of COVID-19. Specifically, we expected that participants who had high trust in these sources might exhibit lower knowledge regardless of their consumption of different news media.

## Materials and methods

### Participants

The study sample included 194 adolescents and young adults ($M_{age}$ = 17.40, $SD$ = 1.74 years, 74.6% female and 25.4% male) from the UK ($n$ = 105) and the US ($n$ = 89). The participants were part of a larger longitudinal study on adolescent learning and STEM engagement in informal STEM learning settings. All respondents participated in STEM-oriented youth programs at informal learning sites in the US and UK. Participant ethnicities were as follows; 36.5% White British or European American, 27.3% Asian British or Asian American, 8.8% Black British or African American, 8.2% Bi-racial, 1.5% Hispanic or Latino, 10.8% selecting "other" and 8.2% choosing not to report their ethnicity. A power analysis conducted with G*Power indicated that a sample size of 110 or greater would enable us to detect small effects ($f^2$ = .10) $\alpha$ = .05, and a power of = .95 for multiple regression analyses with seven predictors (see data analytic plan below).

### Procedure

This research was approved by the Institutional Review Boards at North Carolina State University in the US and University of Exeter in the UK. Adolescents and young adults, who were part of a collaborative longitudinal study between scholars in the US and the UK examining their participation in informal STEM learning programs and changes in their STEM career interests over time, were invited to participate in a survey developed for this study assessing their experiences during COVID-19 in May and June of 2020 [61]. For US participants, opt-out informed consent forms were emailed to the parents of potential participants one week before the survey was emailed to participants. For participants in the UK who were under 16, parents had to opt-in for their child to participate in the study. Participants under 16 whose parents opted-in for their child to participate were included in the study. All youth over the age of 16 were eligible to give their own consent. All participants who had parental consent or did not need consent given their age were emailed an invitation to participate in the study. All participants assented to participation and completed the survey using Qualtrics on their own. Participants were compensated with a small electronic gift card. The survey included items about how students were staying engaged with their informal STEM learning program, but also included items about where students were learning about COVID-19, how much they trusted different sources of knowledge, and what they knew about COVID-19, which were the focus of this analysis.

### Measures

**Frequency of COVID-19 information in news and social media.** Eight items that were developed for this study assessed how often participants encountered COVID-19 information from print, online, and social media outlets (item stem: How often do you encounter COVID-19 information from each of the following sources?). Participants responded to Likert-type

scale items (1 = *Never* to 7 = *Every day*) to assess the frequency of use for each outlet. A combined average score of online and print news media use was calculated to assess traditional news media use ($M = 3.88$, $SD = 1.68$). For social media news, an average score was created of all the listed social media sources ($M = 3.58$, $SD = 1.53$, $\alpha = .71$).

**Trust in source of COVID-19 information.** Six items developed for this study assessed level of trust in various sources of COVID-19 information ($\alpha = .70$). Trust in teachers, government leader, government, the World Health Organization (WHO), news media, and social media were assessed by items that read, "How much do you trust the information presented by the [information source] about COVID-19? Responses were on a 7-point Likert-type ranging from 1 = Don't trust at all to 7 = Trust completely.

**COVID-19 health literacy.** Four factual COVID-19 questions were created for this survey to assess participants' knowledge of COVID-19. Item responses were multiple choice, with one correct response out of four options. Items include "What is "COVID-19?", "Which of the following is true of COVID-19?", "Which of the following is true about viruses?", and "Which of the following is true about this graph?" (see S1 File). Responses for each item were dummy coded (0 = incorrect, 1 = correct) and summed across items to create a total score of COVID-19 questions correct.

**Demographics.** We included gender and country as dummy coded covariates in the model (0 = male, 1 = female for gender and 0 = UK, 1 = US for country). Age was also assessed as a control variable, but was not associated with the variables of interest and, thus, was dropped after preliminary analyses.

**Data analysis.** All analyses for this study were conducted using IBM© SPSS© software version 27. First, we conducted descriptive analyses with study variables (means, standard deviations, and bivariate correlations (Table 1). As preliminary analyses suggested that only trust in government leaders was associated with the number of questions participants got correct, all other trust variables were dropped from analyses. Using hierarchical linear regression [62] we examined direct associations between average traditional news media and social news media, with the number of COVID questions correct. In Step 1, we entered gender and country as

**Table 1. Means, standard deviations, and bivariate correlations for study variables.**

| Variable Name | 1 | 2 | 3 | 4 | 5 | 6 | 7 | 8 | 9 | 10 | 11 | 12 |
|---|---|---|---|---|---|---|---|---|---|---|---|---|
| 1. Country | - | | | | | | | | | | | |
| 2. Gender | -.15* | - | | | | | | | | | | |
| 3. Trust in Teachers | -.03 | -.11 | - | | | | | | | | | |
| 4. Trust in Government Leader | -.16* | -.10 | .29** | - | | | | | | | | |
| 5. Trust in Government | -.03 | -.10 | .23** | .53** | - | | | | | | | |
| 6. Trust in WHO | .05 | .02 | .13 | -.09 | .31** | - | | | | | | |
| 7. Trust in News Media | .11 | -.14 | .25** | .13 | .46** | .41** | - | | | | | |
| 8. Trust in Social Media | .04 | .00 | .23** | .15* | .29** | .34** | .49** | - | | | | |
| 9. Trust in Scientists | .36** | -.11 | .17* | .06 | .25** | .38** | .37** | .16* | - | | | |
| 10. Traditional News Media | -.17* | .05 | .13 | .23** | -.03 | -.05 | .06 | .01 | .02 | - | | |
| 11. Social News Media | -.10 | .11 | .08 | .14 | .01 | .20** | .09 | .29** | .08 | .40** | - | |
| 12. Total COVID-19 Questions Correct | -.25** | -.04 | -.07 | -.17* | .02 | .11 | .07 | .04 | -.09 | -.08 | -.10 | - |
| M | .48 | .76 | 4.76 | 3.34 | 3.92 | 5.40 | 4.03 | 2.98 | 5.44 | 3.88 | 3.58 | 2.21 |
| SD | .50 | .43 | 1.55 | 1.78 | 1.56 | 1.59 | 1.43 | 1.43 | 1.37 | 1.67 | 1.53 | .83 |

**p < .01

*p < .05

our control variables. In Step 2, we entered average traditional news media and social media news as our primary variables of interest. Trust in government leaders' knowledge of COVID-19 was entered at Step 3. Subsequently we examined interactions between primary study variables and trust in government leaders' knowledge of COVID-19. This stepwise approach allowed us to examine the unique and overlapping contributions of the different news sources adolescents used to understand COVID-19. Further examining the interactions provides support for how contextual factors, such as government leadership during the COVID-19 pandemic, may impact the ways that youth interpret the various news sources with which they engage.

## Results

### Preliminary analyses

We examined means, standard deviations, and bivariate correlations among study variables (Table 1). We also assessed the percentage of adolescents who reported using each platform as their news source. On average adolescents had nearly equal use of traditional versus social news media use ($M$ = 3.88, $SD$ = 1.68; $M$ = 3.58, $SD$ = 1.57 respectively). The most frequently used forms of social media were YouTube (89% of the sample) and Instagram (77% of the sample; Table 2). Adolescents exhibited some gaps in their knowledge about COVID-19, averaging 2.21 ($SD$ = .83) questions correct out of four possible questions. Youth had generally low trust in their respective government leaders ($M$ = 3.34, $SD$ = 1.78), and the highest levels of trust in both scientists ($M$ = 5.44) and World Health Organization (WHO; $M$ = 5.40). Bivariate correlations indicated that trust in government leaders about COVID-19 was positively associated with traditional news media. Trust in social media and trust in WHO were positively associated with social media news. Only trust in government leaders was associated (negatively) with the total number of COVID questions correct. As noted, given the lack of association with the other trust variables with total number of COVID questions correct, they were dropped from subsequent analyses.

### News source as predictor of COVID-19 health literacy

We conducted hierarchical linear regression models to examine different types of news sources' association with COVID-19 health literacy (Table 3). In step one, gender and country were entered as control variables. Country was significant, with students from the US getting fewer COVID-19 questions correct. In Step 2, neither traditional media news nor social media use was associated with number of COVID-19 questions correct ($\beta$ = -.24 p = .44; $\beta$ = -.06 p = .21). In Step 3, only trust in government leadership was associated with number of COVID

**Table 2. Percentage of adolescents use of news platforms.**

| News Source | % of Sample Using Source |
|---|:---:|
| Print News | 51 |
| Online News | 90 |
| Facebook | 45 |
| Twitter | 61 |
| Instagram | 77 |
| Tik Tok | 58 |
| Snapchat | 62 |
| YouTube | 89 |

**Table 3. Hierarchical linear regression of news source predicting total number of questions correct.**

| Variable | B | SE B | β | R² | R² change |
|---|---|---|---|---|---|
| Step 1 | | | | .05 | |
| Gender | -0.1 | .14 | -.06** | | |
| Country | -.35 | .12 | -.22** | | |
| Step 2 | | | | .07 | .02 |
| Traditional Media | -.03 | .40 | -.06** | | |
| Social Media | -.05 | .04 | -.10** | | |
| Step 3 | | | | .10 | .03 |
| Trust in Government Leadership | -.09 | .04 | -.18** | | |
| Step 4 | | | | .13 | .03 |
| Trust in Government Leadership X Traditional Media | -.05 | .02 | -.20** | | |
| Trust in Government Leadership X Social Media | .01 | .02 | .40** | | |

**p < .01

*p < .05

questions correct such that those who had higher trust in their government leader's knowledge of COVID-19 got fewer COVID-19 questions correct β = -.18 p < .05.

## Trust in government leadership's knowledge of COVID-19

Interaction terms were entered at step 4 of the hierarchical regression model to examine trust in government leadership's knowledge of COVID-19 (i.e., trust in the COVID-19 information shared by the US President or UK Prime Minister) as a moderator of the association between traditional news media and social news media with the total number of COVID-19 questions correct. Traditional and social news media variables were centered in order to compute the interaction terms. Trust in government leadership moderated the association between traditional news media and total number of COVID-19 questions correct. We conducted simple slopes analyses to probe the interaction (see Fig 1). Simple slopes analyses suggest that respondents with the highest levels of trust in their government leader (1 SD above the mean) got fewer questions correct when they encountered more COVID-19 information in traditional media (β = -.11 p < .05). There was no association between frequency of COVID-19 information in traditional news media and total number of questions correct for those who had average or low levels of trust in their government leader's knowledge of COVID-19.

## Discussion

The present study examined associations between adolescents' news sources and their COVID-19 health literacy. Further, it examined the role of trust in different sources of information, finding that trust in government leaders was central to the association between adolescent news source and COVID-19 health literacy. Moreover, adolescents' literacy was not associated with frequency of accessing news using traditional or social media sources. However, trust in government leaders moderated this effect: for participants who have high trust in government leadership, the more they consume traditional news media, the less accurate they were in their responses to COVID-19 literacy assessment. These findings provide novel insight into how adolescents are encountering information about the pandemic and the importance of considering not just where they are getting their knowledge, but also their trust in government leadership.

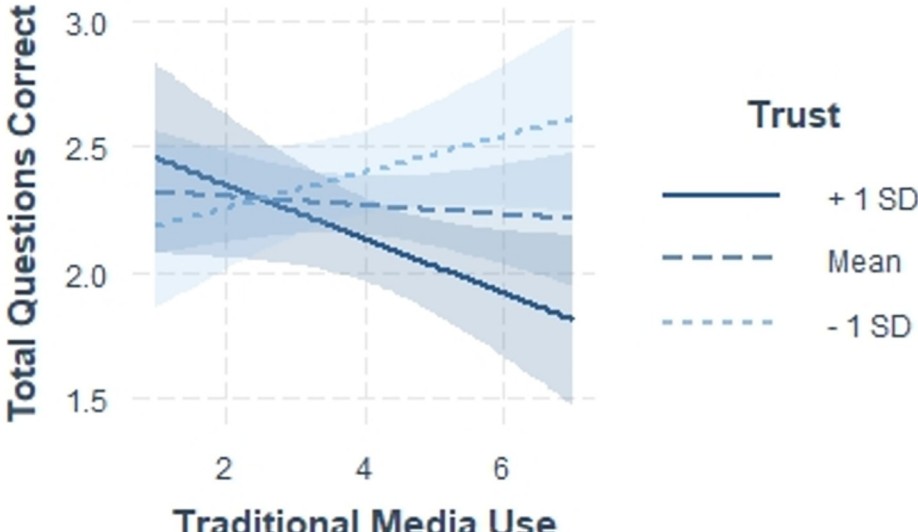

**Fig 1. Relationship between traditional news media and COVID-19 questions correct by trust.** This figure depicts how trust in government leaders moderates the relationship between traditional news media consumption (1 = Never to 7 = Every day) and the number of COVID-19 questions correct. Trust = Trust in Government Leaders' knowledge of COVID-19.

## Adolescents' understanding of COVID-19

Scant prior research has examined adolescents' knowledge of viruses and infectious disease [19, 52]. However, this prior research has documented that adolescents may have incomplete or faulty understanding of viruses and virology. Accurate understanding of infectious disease is a key factor associated with reducing spread and ensuring hygiene habits to curb likelihood of infection, with findings from Australian adults suggesting that health literacy is particularly important for behaviors during the COVID-19 pandemic [63]. Whereas prior research with adolescents in Jordan suggests that adolescents have an adequate understanding of COVID-19 [20], our findings suggest that American and British adolescents may have only average knowledge around COVID-19, with participants, on average only getting about half of the questions correct. At a glance these findings are concerning, as they suggest that adolescents may misunderstand fundamentals of health literacy. Unfortunately, this lack of basic understanding of virology is not surprising: prior research documented that 0% of Canadian high school biology students surveyed were able to provide a scientific definition of the structure and function of a virus in a pretest assessment administered before an intervention to promote knowledge of influenza, although their ability to provide a definition did improve following the intervention [19]. However, a promising finding was that youth performed very well on questions related to public health behavior, particularly being aware of how the virus was transmitted and that vaccines can prevent future outbreaks. Youth's awareness of behaviors that both spread and contain virus transmission has important implications for positive health behaviors during the pandemic. Even so, adolescents' incomplete health literacy is of concern and should be a priority both for public health campaigns as well as for curricular decisions. In particular, youth exhibited greater difficulty with a data science question that required that they interpret a graph and with an item on knowledge of virology. This may suggest the importance of continued attention to teaching the basics of virology and of data science explicitly. While adolescents' knowledge was somewhat average given the STEM-oriented sample, it is worth

highlighting that students completed this survey early in the pandemic (in May of 2020), when most students were not in school or were remote learning. Had students been able to continue to consistently attend school and engage with their teachers, they may have learned more about COVID-19, although prior research on knowledge of HIV indicates that adolescents garnered their information about viruses from sources other than school teachers/curriculum [54–56]. We did ask students an open-ended question about what their teachers were telling them about COVID-19 and over 60 participants (about 31% of the sample) indicated that they were hearing nothing about COVID-19 from their teachers, because the teachers were not teaching about it, they were not in school, or school had been interrupted drastically.

### Adolescents' COVID-19 news sources

We were also interested in assessing where STEM-oriented adolescents were encountering information about COVID-19. Participants reported frequently obtaining information about COVID-19 through social media, with over half of the sample using at least four different social media platforms to access news. YouTube and Instagram had the highest frequency of use, which aligns with previous research suggesting that adolescents prefer visual and video platforms to access news media [28]. Further youth encountered information from multiple social media sources, suggesting that they used multiple platforms to triangulate information and potentially evaluate credibility of the news source. Though research suggests that youth are poor evaluators of real versus fake news [11, 29], the fact that youth are engaged across multiple platforms may suggest a desire to consume multiple perspectives about COVID-19 and other newsworthy events. Such findings are promising given that engaging with more diverse perspectives has a strong association with higher critical thinking, which is an important skill to have when distinguishing fact from fiction [29]. Even if youth have not quite mastered how to evaluate credible news sources, their willingness to engage multiple sources is a step to establishing greater online health literacy, which is important for scientists and public health officials to consider regarding how teens are translating and implementing consumed information about COVID-19.

### Associations between sources of COVID-19 information and knowledge of COVID-19

A key aim of the current study was to examine associations between adolescents' knowledge of COVID-19 and where they encounter information. Though we expected that adolescents who reported encountering more information about COVID-19 on social media would exhibit lower knowledge of COVID-19, due to the high amount of health-related misinformation shared through social media [64], in fact, in our sample, frequency of encountering information about COVID-19 on social media was not associated with accurate knowledge of COVID-19. This may be because adolescents and young adults are increasingly aware of the dangers of misinformation and "fake news" on social media and that young people are increasingly looking with a critical eye at news they encounter on social media platforms [28, 30]. Additionally our sample was made up of STEM-oriented participants that are all part of programming at informal educational institutions–i.e., participants with exposure to STEM-knowledgeable authorities and a propensity to seek STEM educational opportunities. Such exposures may lead them to have higher levels of health literacy than the average adolescent population, given their experiences in informal STEM learning contexts.

Similarly, we expected that adolescents' exposure to COVID-19 related information through traditional news would be associated with greater accuracy on the COVID-19 assessment. However, we found no association. Future research may need to examine exposure to

specific traditional news outlets, as it may be that some provide more or less scientifically accurate health information. Additionally, future research may need to explicitly consider the role of news media paywalls in order to better understand where adolescents are learning about viruses. Increasingly, reputable news organizations do have online coverage, but these news stories may be secured behind paywalls. However, especially early in the pandemic, many journalistic organizations, such as the New York Times, did remove the paywall for articles related to the pandemic in order to ensure that accurate journalism surrounding COVID-19 was accessible.

## Adolescents' trust

Interestingly, although we assessed trust in many different sources, the only source that was associated with COVID-19 knowledge was trust in government leaders. Though we did not find a direct association between the frequency of COVID-19 information on traditional news media or social news media and COVID-19 knowledge, we did find a moderating effect of trust in government leaders COVID-19 knowledge for youth who reported using traditional news media. For those with low and average trust in government leadership, there was no relation between exposure to traditional news media and knowledge of COVID-19. However, for American and British adolescents with greater trust in government officials, the more exposure to COVID-19 information on traditional media adolescents reported, the lower they scored on the COVID-19 assessment. This finding counters previous work with adult samples that suggests higher trust in government information sources is associated with greater accuracy in COVID-19 knowledge [14]. Such findings suggest that the messaging of government leaders is critical, and must be aligned with public health recommendations. The findings document that government leaders likely have an impact on adolescents' knowledge. Government officials who provide timely and scientifically accurate information may have a positive impact by guiding youth and adult populations alike to credible scientific information to slow the spread of COVID-19.

## Limitations and future directions

There are a few limitations in this study. First, due to the cross-sectional nature of the study, we only provide a snapshot (albeit of the critical May-June 2020 time period near the first explosive waves of the pandemic) of potential associations between adolescents' news consumption and knowledge of COVID-19. Given the rapidly changing information regarding the structure, transmission, and prevention of the virus, it is imperative that researchers examine how frequency of exposure to COVID-19 news over time may increase or decrease their knowledge of COVID-19. Scholars emphasize that we are navigating an 'infodemic' amidst a pandemic due to the amount of misinformation across media platforms [2, 6, 64]. Thus, understanding how adolescents and young adults consume and interpret news is critical to identifying ways to help youth access and interpret credible information about COVID-19 and other global public health issues.

Finally, the sample for which this study was conducted is likely not broadly representative of adolescent youth across the US and the UK given that the participants were approximately 75% female and were explicitly drawn from a larger longitudinal study focused on students' participation in informal science learning programs. Thus, these adolescents likely have a higher interest in STEM knowledge than the general population. Though future research should assess more nationally representative samples across the US and the UK to examine if frequency of COVID-19 news and knowledge of COVID-19 varies across more general adolescent populations, the STEM-oriented participants in our current sample could be future

leaders in the fight against the next pandemics, and therefore understanding their knowledge and the sources of that knowledge is important.

## Implications

Youth's COVID-19 knowledge has implications for their future behavior, such as adherence to social distancing guidelines, or wearing masks in public settings. In addition, youth equipped with more knowledge about COVID-19 may be more inclined to share information that encourages others to develop stronger health literacy around COVID-19 and other public health crises. Given that youth are more likely to use social media as a tool for keeping themselves informed about global issues, it may also be key to facilitating such awareness in others [65, 66]. Such critical awareness in youth may translate beyond understanding the COVID-19 virus to engaging the social impact of COVID-19 particularly when it comes to health disparities. In the summer of 2020, protests erupted across the US as Black, Latinx, and poor communities struggled for access to COVID-19 testing and treatment facilities [67, 68]. Such protests underscored the social burden that many marginalized groups endure, particularly those who experience multiple intersections of marginalization [67, 68]. Such protests reignited discussions of racial disparities in access to healthcare and treatment that permeate the nation's history. By remaining critically informed of COVID-19 and their roles in slowing the spread of COVID-19, adolescents may be a key factor in helping to reduce disproportionate impacts of COVID-19 by bringing awareness to how the consequences of viral infection vary across demographic contexts.

## Conclusion

The present study examined if exposure to COVID-19 information across various media platforms were associated with their COVID-19 health literacy. Further, this research considered the role of trust in different sources of information, including trust in government leaders' information, in the association between frequency of COVID-19 information from news outlets and COVID-19 knowledge. The findings highlight that trust in government leaders plays a critical role in COVID-19 public health messaging for adolescents, and such messaging may not consistently translate into greater accuracy factual knowledge of COVID-19. The necessity of understanding adolescents' sources of COVID-19 knowledge cannot be underestimated given the implications of how such knowledge informs their social distancing and hygiene behaviors. Understanding the ways in which youth consume, interpret, and translate COVID-19 news is a necessary step to slowing the spread of COVID-19, and bring an end to the crisis that has taken hundreds of thousands of lives across the globe.

## Supporting information

**S1 File. Scale items.**
(DOCX)

## Author Contributions

**Conceptualization:** Channing J. Mathews, Luke McGuire, Mark Winterbottom, Adam Rutland, Adam J. Hoffman, Kelly Lynn Mulvey, Adam Hartstone-Rose.

**Data curation:** Channing J. Mathews, Luke McGuire, Adam J. Hoffman, Kelly Lynn Mulvey, Adam Hartstone-Rose.

**Formal analysis:** Channing J. Mathews, Kelly Lynn Mulvey.

**Funding acquisition:** Mark Winterbottom, Adam Rutland, Kelly Lynn Mulvey, Adam Hartstone-Rose.

**Investigation:** Kelly Lynn Mulvey, Adam Hartstone-Rose.

**Methodology:** Channing J. Mathews, Kelly Lynn Mulvey, Adam Hartstone-Rose.

**Project administration:** Luke McGuire, Adam Rutland, Adam J. Hoffman, Kelly Lynn Mulvey.

**Resources:** Kelly Lynn Mulvey, Adam Hartstone-Rose.

**Software:** Channing J. Mathews, Kelly Lynn Mulvey, Adam Hartstone-Rose.

**Supervision:** Kelly Lynn Mulvey, Adam Hartstone-Rose.

**Visualization:** Kelly Lynn Mulvey.

**Writing – original draft:** Channing J. Mathews, Kelly Lynn Mulvey.

**Writing – review & editing:** Channing J. Mathews, Luke McGuire, Angelina Joy, Fidelia Law, Mark Winterbottom, Adam Rutland, Marc Drews, Adam J. Hoffman, Kelly Lynn Mulvey, Adam Hartstone-Rose.

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
