## [Decision Letter · Decision Letter 0]

19 Aug 2021

PONE-D-21-12139

Assessing adolescents’ critical health literacy: How is trust in government leadership associated with knowledge of COVID-19?

PLOS ONE

Dear Dr. Matthews,

Thank you for submitting your manuscript to PLOS ONE. After careful consideration, we feel that it has merit but does not fully meet PLOS ONE’s publication criteria as it currently stands. Therefore, we invite you to submit a revised version of the manuscript that addresses the points raised during the review process.

Please submit your revised manuscript within 60 days of receipt of this email. If you will need more time than this to complete your revisions, please reply to this message or contact the journal office at plosone@plos.org. Please include the following items when submitting your revised manuscript:

We look forward to receiving your revised manuscript.

Kind regards,

Marlene Camacho-Rivera, ScD, MPH

Academic Editor

PLOS ONE

Journal Requirements:

3. In order to improve reporting, in your methods section, please provide additional information about the participant recruitment method and the demographic details of your participants, and in particular provide additional information about the broader project they were recruited for.

4. Peer review at PLOS ONE is not double-blinded (https://journals.plos.org/plosone/s/editorial-and-peer-review-process). For this reason, authors should include in the revised manuscript all the information removed for blind review.

Reviewers' comments:

Reviewer's Responses to Questions

**Comments to the Author**

1. Is the manuscript technically sound, and do the data support the conclusions?

Reviewer #1: Yes

2. Has the statistical analysis been performed appropriately and rigorously? 

Reviewer #1: Yes

3. Have the authors made all data underlying the findings in their manuscript fully available?

Reviewer #1: Yes

4. Is the manuscript presented in an intelligible fashion and written in standard English?

Reviewer #1: Yes

5. Review Comments to the Author

Reviewer #1: 1. This present study give knowledge to readers about the associations of health literacy and how is trust in government leadership associated with knowledge of COVID-19. This is interesting study that expanding the health literacy in the context of crucial situation like pandemic COVID19. However there are some justification needs to be done.

a) How many overall population of STEM students for both countries ?

b) the authors did mention the limitation of study intern of sampling. The sample size is 194 consisted 2 categories, Does this sample size is enough for hierarchical linear regression?

c) What is the effect size of the moderation analysis?

d) How about the validation of items measurement ?

It would be good if the authors spell out all items measured in the appendix.

6. PLOS authors have the option to publish the peer review history of their article (what does this mean?). If published, this will include your full peer review and any attached files.

Reviewer #1: **Yes: **Mohammad Rezal Hamzah

---

## [Author Response · Author response to Decision Letter 0]

1 Sep 2021

These responses are detailed in our response to reviewers letter, but are also included here as per the instructions:

Dear Dr. Camacho-Rivera,

Thank you for your careful feedback to our manuscript entitled: Assessing adolescents’ critical health literacy: How is trust in government leadership associated with knowledge of COVID-19? We appreciate and have addressed your overall comments as well as comments from our reviewer, Dr. Hazmah, throughout our last manuscript draft. In this cover letter, the exact language from the action letter is first quoted in italics, followed by our response in plain text. Minor changes, such as updating existing citations and style requirements, are not detailed here, but are shown in the revised manuscript with tracked changes.

1.) You indicated that you had ethical approval for your study. In your Methods section, please ensure you have also stated whether you obtained consent from parents or guardians of the minors included in the study or whether the research ethics committee or IRB specifically waived the need for their consent.

Thank you for this feedback. We clarified in the methods section that we obtained consent from the parents or guardians of minors included in the study from both countries. 

2.) In order to improve reporting, in your methods section, please provide additional information about the participant recruitment method and the demographic details of your participants, and in particular provide additional information about the broader project they were recruited for.

Thank you for this suggestion. We have included more descriptive information to highlight the international collaboration between the US and UK and described the goal of the original project to assess trajectories adolescents’ science career interests over time. 

3.) In your Data Availability statement, you have not specified where the minimal data set underlying the results described in your manuscript can be found.

Thank you for pointing out this error. The dataset was previously included in the supplemental files but has now been deposited on the Open Science Framework. It can be found at the following URL: https://osf.io/nj3bf/

4.) How many overall population of STEM students for both countries? 

The overall population of STEM students is 194. There are 105 in the UK and 89 in the US. This information is indicated in the Participants section on pg. 14.

5.) The authors did mention the limitation of study intern of sampling. The sample size is 194 consisted of 2 categories, does this sample size is enough for hierarchical linear regression?

Thank you for this question. We included a power analysis using G*Power that is described on page 14 of the manuscript. Power analysis indicated that a sample size of greater than 110 would enable us to detect a small effect. 

6.) What is the effect size of the moderation analysis?

The effect size of the moderation analysis is indicated by the R2 value indicated in Table 3 on page 20 of the revised manuscript. R2=.13

7.) How about the validation of items measurement?

We added wording to indicate that all the measures were newly developed for this survey. We also included alpha levels to indicate the reliability of applicable measures. 

8.) It would be good if the authors spell out all items measured in the appendix.

Thank you for this suggestion. We have updated the appendix to include descriptions of all the items measured. 

Thank you for your review of our manuscript. We are excited to move forward in the next steps of the revision process. 

Sincerely and on behalf of all co-authors,

Channing J. Mathews, PhD

Postdoctoral Research Fellow, Social Development Lab

North Carolina State University

---

## [Decision Letter · Decision Letter 1]

21 Oct 2021

Assessing adolescents’ critical health literacy: How is trust in government leadership associated with knowledge of COVID-19?

PONE-D-21-12139R1

Dear Dr. Matthews,

We’re pleased to inform you that your manuscript has been judged scientifically suitable for publication and will be formally accepted for publication once it meets all outstanding technical requirements.

Kind regards,

Marlene Camacho-Rivera, ScD, MPH

Academic Editor

PLOS ONE

Additional Editor Comments (optional):

Reviewers' comments:

Reviewer's Responses to Questions

**Comments to the Author**

1. If the authors have adequately addressed your comments raised in a previous round of review and you feel that this manuscript is now acceptable for publication, you may indicate that here to bypass the “Comments to the Author” section, enter your conflict of interest statement in the “Confidential to Editor” section, and submit your "Accept" recommendation.

Reviewer #1: All comments have been addressed

2. Is the manuscript technically sound, and do the data support the conclusions?

Reviewer #1: Yes

3. Has the statistical analysis been performed appropriately and rigorously? 

Reviewer #1: Yes

4. Have the authors made all data underlying the findings in their manuscript fully available?

Reviewer #1: Yes

5. Is the manuscript presented in an intelligible fashion and written in standard English?

Reviewer #1: Yes

6. Review Comments to the Author

Reviewer #1: (No Response)

7. PLOS authors have the option to publish the peer review history of their article (what does this mean?). If published, this will include your full peer review and any attached files.

Reviewer #1: **Yes: **Mohammad Rezal Hamzah

---

## [Editor Report · Acceptance letter]

25 Oct 2021

PONE-D-21-12139R1 

Assessing adolescents’ critical health literacy: How is trust in government leadership associated with knowledge of COVID-19? 

Dear Dr. Mathews:

I'm pleased to inform you that your manuscript has been deemed suitable for publication in PLOS ONE. Congratulations! Your manuscript is now with our production department. 

Kind regards, 

on behalf of

Dr. Marlene Camacho-Rivera 

Academic Editor

PLOS ONE